# Type I Diabetes in Zebrafish Reduces Sperm Quality and Increases Insulin and Glucose Transporter Transcripts

**DOI:** 10.3390/ijms24087035

**Published:** 2023-04-11

**Authors:** Patrícia Diogo, Gil Martins, Márcio Simão, Ana Marreiros, Ana Catarina Eufrásio, Elsa Cabrita, Paulo Jorge Gavaia

**Affiliations:** 1Faculty of Sciences and Technology (FCT), University of Algarve, 8005-139 Faro, Portugal; gsmartins@ualg.pt (G.M.);; 2Centre of Marine Sciences (CCMAR), University of Algarve, 8005-139 Faro, Portugal; 3Necton-Companhia Portuguesa de Culturas Marinhas S.A, Belamandil s/n, 8700-152 Olhão, Portugal; 4Faculty of Medicine and Biomedical Sciences (FMCB), University of Algarve, 8005-139 Faro, Portugal; ammarreiros@ualg.pt; 5Algarve Biomedical Center (ABC), University of Algarve, 8005-139 Faro, Portugal; 6Instituto de Investigação e Inovação em Saúde (i3S), Universidade do Porto, 4200-135 Porto, Portugal; 7Instituto de Biologia Molecular e Celular (IBMC), 4200-135 Porto, Portugal

**Keywords:** zebrafish, type I diabetes, sperm motility, DNA integrity

## Abstract

Type I diabetes is a prominent human pathology with increasing incidence in the population; however, its cause is still unknown. This disease promotes detrimental effects on reproduction, such as lower sperm motility and DNA integrity. Hence, the investigation of the underlying mechanisms of this metabolic disturbance in reproduction and its transgenerational consequences is of the utmost importance. The zebrafish is a useful model for this research considering its high homology with human genes as well as its fast generation and regeneration abilities. Therefore, we aimed to investigate sperm quality and genes relevant to diabetes in the spermatozoa of Tg(*ins*:*nfsb-mCherry*) zebrafish, a model for type I diabetes. Diabetic Tg(*ins*:*nfsb-mCherry*) males showed significantly higher expression of transcripts for *insulin a* (*insa*) and *glucose transporter* (*slc2a2*) compared to controls. Sperm obtained from the same treatment group showed significantly lower sperm motility, plasma membrane viability, and DNA integrity compared to that from the control group. Upon sperm cryopreservation, sperm freezability was reduced, which could be a consequence of poor initial sperm quality. Altogether, the data showed similar detrimental effects related to type I diabetes in zebrafish spermatozoa at the cellular and molecular levels. Therefore, our study validates the zebrafish model for type I diabetes research in germ cells.

## 1. Introduction

Diabetes mellitus is a metabolic disorder characterized by chronic hyperglycemia with disturbances of carbohydrate, fat, and protein metabolism. This metabolic unbalance results from defects in insulin secretion and/or action [1]. It is estimated that 382 million people suffer from diabetes, with 8.2% prevalence [2]. The projection for the human population suffering from diabetes in 2035 is expected to reach up to 592 million people [2]. Type I diabetes is a chronic autoimmune disease characterized by the loss of insulin-producing β cells in the pancreas, leading to insulin deficiency. Type II diabetes is the acquired insulin resistance, which can occur in combination with reduced insulin secretion [3]. Both types of diabetes show a growing incidence among the human population. The cause of type II diabetes is known to be primarily lifestyle factors and genetic predisposition [4,5]. The cause of type I diabetes is polygenic; the threshold hypothesis by Atkinson and colleagues considers that its onset does not occur if the combined effects of environmental and genetic factors do not exceed the disease threshold [6].

Considering the alarming increase in diabetes incidence and the fact that environmental factors affect both maternal and paternal germ cells, there is a great need for intergenerational and transgenerational studies of type I diabetes [7] on offspring inheritance and reproductive function. For this purpose, a model species with fast generation is necessary. The zebrafish is a promising candidate for this type of research, since it is a small-bodied teleost with fast generation, the genome is fully sequenced, and it has 74% homology with human genes [8]. Additionally, there are established transgenic zebrafish models for type I diabetes [9,10]. This species has high regeneration ability and is therefore able to regenerate ablated pancreatic β cells [11], which makes it a particularly useful model for the investigation of type I diabetes.

Among other complications, diabetes causes disturbances in the male reproductive system since glucose metabolism is an important event not only in spermatogenesis [7] but also in mature spermatozoa metabolism [12,13]. When critical developmental points of spermatogenesis are affected, environmentally induced epigenetic modifications may become permanent in the germ line epigenome, with potential consequences on subsequent generations [7,14]. Numerous studies have been performed both in humans and mouse models, confirming the deleterious effects of diabetes on reproduction and gamete quality [7,15]. In males, these deleterious consequences are particularly evident in sperm quality parameters such as DNA fragmentation, chromatin quality, sperm motility, and seminal plasma composition [16]. These deleterious effects on reproduction are observed both in type I and type II diabetes [7] and even in pre-diabetic and obesity conditions [17,18].

Sperm cryopreservation is not only a valuable resource to support assisted reproduction, but it is also an indicator of the cells’ susceptibility to cold exposure or freezability [19,20]. In humans with type I diabetes, sperm cryopreservation can be a valuable tool to safeguard the possibility of in vitro fertilization later in life [21]. In zebrafish, sperm cryopreservation is an important tool to back-up valuable zebrafish genetic resources that support biomedical research. Factors affecting the plasma membrane composition and fluidity [19], sperm subpopulations structure [22], intrinsic male variability [23,24], and differences in the abundance of proteins relevant to sperm function [23] are associated with sperm freezability. However, sperm freezability predictors are not universally manifested in sperm quality traits across species prior to cryopreservation [24]. Recently, good freezability of carp (*Cyprinus carpio*) sperm was related to high concentrations of proteins responsible for the maintenance of flagella structure, membrane fluidity, sperm motility, and energy production, which can be markers of spermatozoa full maturation [25].

Unexpectedly, it was demonstrated that mammalian spermatozoa have stores of insulin [26,27]. Moreover, spermatozoa secrete this hormone in a short autocrine loop to recruit glucose as an energy substrate [28]. This recruitment of glucose through insulin secretion is performed according to metabolic needs or alterations in systemic energy homeostasis [28]. The presence of insulin in teleost sperm still needs to be investigated. However, if similar to mammals, the role of insulin can be relevant not only in somatic cells, but also in germ cells. Spermatozoa require sugar uptake for energy production and motility [29] through active or passive glucose transporters in the membrane (GLUTs) [30]. Considering the fact that diabetes promotes alterations in sugar metabolism, it can potentially affect the genetic expression of sugar transporters in spermatozoa. Additionally, diabetes is known to increase oxidative stress, which is deleterious to spermatozoa function. Therefore, it is pertinent to understand the alterations in transcripts of *insulin a* (*insa*), *insulin receptor a* (*insra*), and *glucose carrier* (*slc2a2*) under transient diabetic conditions, which will be inherited by the offspring.

The objective of this study was to investigate the sperm quality of a zebrafish transgenic model of type I diabetes, Tg(*ins*:*nfsb-mCherry*), under transient diabetic conditions. The evaluation of target genes relevant for research of type I diabetes, such as *insa*, *insra*, and *slc2a2*, was performed. Additionally, a preliminary analysis of *1,4- galactosyltransferase* (*b4galt2*), a transcript downregulated in human patients, and genes related to oxidative stress, namely *caspase 3* (*casp3a*) and *hypoxia-inducible factor-1α* (*hif1α*), was performed in zebrafish sperm. These evaluations were performed to obtain a deeper understanding of the usefulness of this model.

## 2. Results

Males were sacrificed by hypothermal shock in an ice water slurry and sperm was immediately collected. To evaluate the diabetic and control conditions of Tg(*ins*:*nfsb-mCherry*) (Figure 1A) after sample collection, Tg(*ins*:*nfsb-mCherry*) males were dissected and pancreatic fluorescence was observed under a fluorescence stereomicroscope. Fish under diabetic conditions (Tg(*ins*:*nfsb-mCherry*) exposed to Met) showed evident pancreatic β-cell ablation (no fluorescent signal) and onset of diabetes (Figure 1B) compared to control individuals (evident fluorescent signal) showing the presence of live β cells (Figure 1C).

In spermatozoa, the transcripts of *insa* and *slc2a2* were present in significantly higher quantities in Tg(*ins*:*nfsb-mCherry*) males under diabetic conditions than in Tg(*ins*:*nfsb-mCherry* untreated controls and in AB males with and without Met exposure (Figure 2A,B). The *insra* transcripts were present in significantly higher quantities in sperm from Tg(*ins*:*nfsb-mCherry*) males under diabetic conditions than in sperm from AB males exposed to Met; however, this was not significantly different to sperm of both lines with the control treatment (Figure 2C). Additionally, *hif1a*, *casp3a* and *b4galt2* expression was evaluated in a preliminary analysis in which it was possible to observe a tendency towards the downregulation of these genes in sperm from Tg(*ins*:*nfsb-mCherry*), particularly under transient diabetic conditions in relation to the controls (AB and Tg(*ins*:*nfsb-mCherry*)) (Appendix A). In fresh sperm, the plasma membrane viability of Tg(*ins*:*nfsb-mCherry*) males under diabetic conditions was significantly reduced compared to those exposed to the control treatment (Figure 3).

The evaluation of sperm motility parameters and DNA fragmentation was performed both in fresh and cryopreserved sperm to evaluate the cellular susceptibility to cold exposure. For sperm motility analysis, a repeated measures ANOVA was initially used to investigate the effect of the treatments (control and diabetic) and sperm cryopreservation throughout post-activation time on sperm motility parameters and their interactions (Table 1). Both treatment and cryopreservation factors showed significant differences on sperm motility parameters without interactions between these factors (Table 1). To thoroughly investigate the effects of each treatment, an independent samples *t*-test was used to study the differences at each time post-activation on fresh and cryopreserved sperm (Table 1). The sperm total motility was significantly reduced in Tg(*ins*:*nfsb-mCherry*) males under diabetic conditions compared to those under control conditions (Figure 4A) in the last seconds of the spermatozoa lifespan both in fresh and cryopreserved sperm (Table 1). There were no significant differences in sperm progressive movement (Figure 4B). The Tg(*ins*:*nfsb-mCherry*) males under diabetic conditions showed significantly reduced sperm velocities and linearity compared to AB males (Figure 4C–E) after 30 s post-activation until the end of their lifespan. This result was observed both in fresh and cryopreserved sperm (Table 1).

The DNA fragmentation was significantly increased by the cryopreservation procedure in sperm from Tg(*ins:nfsb-mCherry*) under both control and diabetic conditions (Figure 5). The Tg(*ins:nfsb-mCherry*) males exposed to diabetic conditions produced significantly higher spermatozoa DNA fragmentation than those exposed to the control treatment (Figure 5).

## 3. Discussion

The zebrafish is a strong candidate to model prominent human pathologies such as diabetes mellitus. The cause of type I diabetes is polygenic and its incidence is growing in the human population, therefore requiring deeper investigation. Diabetes is a metabolic disorder that affects the production of insulin in pancreatic β cells. Amongst the most important complications related to this disease is the impairment of male reproduction, affecting both spermatogenesis and mature sperm metabolism [12]. Diabetes modulates spermatozoa substrate consumption and/or production due to altered glycolytic behavior, with deregulation of glucose uptake and metabolism [12]. Diabetes impairs sperm quality parameters such as motility, DNA integrity, and plasma membrane viability, and it alters seminal plasma composition. This damage observed in spermatozoa from diabetic patients is related to oxidative stress due to the lack of glucose metabolism homeostasis, thus promoting lower embryo quality and early onset of some childhood diseases [7]. The zebrafish has a fast generation cycle, available gene editing tools, and high homology with human genes [8,31]. Moreover, this species has been considered a useful model vertebrate for reproduction [32] and metabolic diseases such as diabetes [33]. Therefore, this model species is a powerful tool for the investigation of this disease’s transgenerational effects and inheritance.

In male type I diabetes patients, spermatogenesis disruption and germ cell apoptosis are observed, which have been associated with local autoimmune damage [34]. Apoptosis is the process of programmed cell death under stressful conditions [35]. Upon activation of the apoptosis cascade, the cell experiences DNA fragmentation, degradation of cytoskeletal and nuclear proteins, crosslinking of proteins, formation of apoptotic bodies, expression of ligands for phagocytic cell receptors, and finally uptake by phagocytic cells [35,36]. In patients with type I diabetes, changes in the expression of genes involved in DNA repair and replication are correlated with increased sperm DNA fragmentation [16]. In our work, it was possible to observe that sperm from zebrafish under diabetic conditions showed significantly lower sperm motility, plasma membrane viability, and DNA integrity compared to sperm from the control treatment group. This data suggested that the ablation of pancreatic β cells and impaired insulin secretion produced germ cell apoptosis, leading to reduced sperm quality and DNA integrity (Figure 6). This study also performed a preliminary analysis of *hif1a* and *casp3a* expression to investigate possible evidence of oxidative stress damage in the spermatozoa of zebrafish under different treatments (Appendix A). Our data showed a tendency towards the reduction in *hif1a* expression under diabetic conditions, and it is known that the repression of hypoxia-inducible factor-1 contribute to increased production of mitochondrial reactive oxygen species in diabetes [37]. Additionally, a preliminary evaluation of *b4galt2* expression in zebrafish sperm from all treatments groups was performed, since this gene is the most downregulated in spermatozoa from human patients [38]. This gene is responsible for biosynthesis and cell membrane components, and the preliminary data obtained showed that *b4galt2* presented a tendency towards downregulation in the spermatozoa of Tg(*ins*:*nfsb-mCherry*) zebrafish, particularly under transient type I diabetes conditions. However, these genes require deeper investigation under these conditions. This observation is in agreement with previous observations in spermatozoa from diabetes patients [38] suggesting that mechanisms associated with membrane protein biosynthesis are affected in spermatozoa. Humans and zebrafish have different reproductive strategies; nevertheless, the metabolic disorder produced by diabetic conditions showed similar detrimental effects on sperm quality.

In zebrafish, stored ATP is considered the basis for motility soon after initiation of motility. However, prolonged motility relies on oxidative phosphorylation and nascent ATP generation [39]. Our results showed that both fresh and cryopreserved sperm had significantly lower motility parameters under diabetic conditions than under control conditions. There was a significant decrease in sperm motility parameters such as VCL, VSL, and LIN, particularly after 30 s post-activation, in sperm from Tg(*ins*:*nfsb-mCherry*) males under diabetic conditions. This result suggested that in Tg(*ins*:*nfsb-mCherry*) males under diabetic conditions, ATP stores allow normal sperm motility in the first seconds of motility, but de novo ATP synthesis through oxidative phosphorylation is impaired in the lasts seconds of motility. The freezability was reduced in sperm from the diabetic treatment group. However, this result could be caused by their lower initial sperm quality and not due to an increase in cellular susceptibility to cold exposure. Further investigation is needed to understand sperm freezability of zebrafish males under diabetic conditions.

Sperm cells are among the most differentiated cells of the organism. They are equipped to leave the body and maintain motion competence to reach the oocyte and perform fertilization. The spermatozoa head is designed to accommodate the paternal DNA and fuse with the oocyte, the mid-piece is dedicated to energy production and homeostasis, and the tail is responsible for its motility [27]. Therefore, motility is one of their most important features, and energy is generated to maintain motion [40]. Spermatozoa need energy to initiate and maintain motility in order to reach the oocyte. This process requires the consumption of adenosine triphosphate (ATP). The metabolic pathways involved in energy production in spermatozoa are anaerobic glycolysis, mitochondrial oxidative phosphorylation, and the pentose phosphate pathway [29]. Sperm primarily use sugars such as glucose, mannose, and fructose as energy fuel for ATP production [29]. These sugars are incorporated passively through lipid bilayers in a slow and inefficient manner and therefore require carriers. Hexoses (sugars) are transported into sperm actively through sodium-dependent glucose transporters (SGLT) or passively through glucose transporters (GLUTs) [30]. GLUTs are essential during passive glucose transport through the blood–testes barrier, which is an important event during spermiogenesis. In addition, GLUTs are present in mature spermatozoa, which require carriers to incorporate energy resources, as previously mentioned. These proteins are located on acrosomal and end pieces of the tail of human spermatozoa [29]. The GLUTs have been considered useful markers of sperm quality for both clinical and commercial purposes [29]. The conditions in which spermatogenesis occurs will imprint genomic alterations, which are inherited by the offspring. The onset of diabetes promotes the impairment of insulin secretion and high systemic glucose levels. Insulin deficiency in human Sertoli cells caused altered glucose consumption, expression of metabolism-associated genes involved in lactate production and export, and adaptation of glucose uptake by modulating the expression of GLUTs [41]. GLUT 2 is a high-affinity glucose transporter and it is expressed at a very high level in pancreatic β cells.

Our data demonstrated that zebrafish sperm under diabetic conditions showed a higher number of transcripts for *glucose carrier 2* (*slc2a2*) than the controls. This data suggested that there is increased importation of intracellular glucose in the spermatozoa of fish under diabetic conditions [42]. In addition, it was possible to detect upregulation of *insa* and *insra* in spermatozoa, which was in agreement with the observations in mammalian spermatozoa [26,27]. To the best of our knowledge, this is the first report of insulin expression in sperm of a fish model; however, future studies should further research on this matter. Under this context, the increase in glucose levels leads to the upregulation of *insa* and *slc2a2* transcripts, as previously observed in another zebrafish model [42]. Similarly, in our study, fish lacking insulin-expressing β cells exhibited high levels of systemic glucose, which consequently promoted a response in spermatozoa cells and led to significant upregulation of *slc2a2* compared to the other treatments. Therefore, our results suggest that the lack of systemic insulin secretion during zebrafish spermatogenesis promotes an increase in systemic glucose levels and consequently increases the expression of *insa* and *slc2a* transcripts in spermatozoa.

In zebrafish, *insra* promotes glycolysis to maintain blood glucose homeostasis and inhibits gluconeogenesis [43]. The diabetes pathology also promotes a deregulation of hormonal balance since insulin is an anabolic hormone. In our study, Tg(*ins*:*nfsb-mCherry*) zebrafish with the transient type I diabetes phenotype exhibited increased levels of *insra* gene transcripts compared to all of the controls (AB zebrafish with and without exposure to the drug and Tg(*ins*:*nfsb-mCherry*) without drug exposure). However, *insra* expression was only significantly higher in sperm from Tg(*ins*:*nfsb-mCherry*) exposed to Met when compared to sperm from AB males exposed to Met. The conditions necessary to expose the fish to the prodrug may have produced stress, thus increasing blood glucose levels and the number of *insra* transcripts in sperm. The data suggested that the systemic absence of insulin stimulated insulin expression in zebrafish sperm to improve glucose uptake under unbalanced systemic blood glucose homeostasis conditions. In our study, we observed that zebrafish sperm increased the number of transcripts of *insa*, *insra* and *slc2a2* under diabetic conditions. This event can be a consequence of cellular transcription during spermatogenesis under diabetic conditions, which will be imprinted in spermatozoa and inherited by the offspring. Altogether, these facts led to a hypothetical model of the putative effects of type I diabetes represented in Figure 6.

The results of this work revealed that detrimental effects are observed on zebrafish sperm viability, motility, and DNA fragmentation under diabetic conditions, similar to those in human patients. The use of zebrafish in diabetes research respects the 3Rs rule since this species can regenerate pancreatic β cells and the diabetic condition can be induced repeatedly on the same individual. The present work provides compelling evidence that the zebrafish is a suitable model for investigating the effects of type I diabetes mellitus on male reproductive function and its transgenerational consequences.

## 4. Materials and Methods

### 4.1. Fish Husbandry

Adult AB and Tg(*ins*:*nfsb-mCherry*) (8–12 months old) zebrafish males were selected according to similar size and maintained in 3.5-L tanks with 15 fish each. The wild-type AB line was provided by the Max Planck Institute for Heart and Lung Research (Bad Nauheim, Germany) and maintained at the Centre of Marine Sciences (CCMAR, Portugal). The reporter line Tg(*ins*:*nfsb-mCherry*) was kindly provided by the Laboratory of Molecular Biology and Genetic Engineering (GIGA Research, Liege, Belgium). The Tg(*ins*:*nfsb-mCherry*) line was previously pre-screened and selected according to the presence of cell signaling in the pancreas during the larval stage. Both fish lines were reared in a ZebTEC^®^ (Tecniplast, Buguggiate, Italy) recirculation system with 980 L of water, as previously described by Diogo et al. [44]. The water system was maintained at 28.2 ± 0.5 °C, 700 ± 75 µS, and pH 7.5 ± 0.2. The fish were fed twice a day with Artemia *nauplii* (AF480, INVE, Dendermonde, Belgium) and ZEBRAFEED^®^ diet (Sparos Lda, Olhão, Portugal) ad libitum.

The Tg(*ins*:*nfsb-mCherry*) zebrafish type I diabetes model was developed by Pisharat et al. [10] with a Tübingen AB background to investigate pancreatic β-cell regeneration. The earliest known marker of β cells in zebrafish embryos is the preproinsulin (*ins*) gene, the promoter of which is expressed in the nascent endocrine pancreas [45]. Thus, a construct was created in which the *nfsb* gene of *Escherichia coli* and the florescent protein mCherry were inserted downstream to the promoter region of the *insa* gene in the Tg(*ins*:*nfsb-mCherry*) line. The expression of the *nfsb* gene produces the nitroreductase (NTR) enzyme, which converts prodrugs such as metronidazole (Met; Sigma-Aldrich, Madrid, Spain) to cytotoxins, resulting in cell apoptosis [10]. Consequently, the Tg(*ins*:*nfsb-mCherry*) line in the presence of Met will convert this prodrug into cytotoxins through the NTR enzyme, therefore ablating the pancreatic β cells and losing mCherry fluorescence in the pancreas.

### 4.2. Experimental Design

Adult AB and Tg(*ins*:*nfsb-mCherry*) zebrafish males (8–12 months) were selected and separated into 2 groups: the control (AB *n* = 7; Tg(*ins*:*nfsb-mCherry*) *n* = 8) and the diabetic treatment (AB *n* = 24; Tg(*ins*:*nfsb-mCherry*) *n* = 21). The Tg(*ins*:*nfsb-mCherry*) males were exposed either to control conditions or exposed to Met to induce transient diabetes. A second Met exposure was performed 7 days after the beginning of the experiment. The sperm was collected three days after the second exposure to Met, when the maximum β-cell ablation occurred [46]. Males were sacrificed by hypothermal shock in an ice slurry at 2 ± 2 °C (monitored with a probe), sperm samples were immediately collected, and the pancreatic fluorescence (Tg(*ins*:*nfsb-mCherry*) line) was evaluated through animal dissection and observation under a fluorescence stereomicroscope. Fish under diabetic conditions showed evident pancreatic β-cell ablation (no fluorescent signal) and onset of diabetes compared to control individuals (evident fluorescent signal) with the presence of live β cells.

To understand the impact of the diabetic conditions on zebrafish spermatozoa at the molecular level, a set of target genes was investigated with high relevance for this metabolic disorder. In fresh sperm, the quantities of i*nsa*, *inra*, and *slc2a2* transcripts were evaluated in both zebrafish lines under control and Met treatment conditions. A preliminary evaluation was performed on *casp3a*, *hif1a* and *b4galt2* transcripts in sperm from all treatments. Plasma membrane viability was evaluated in Tg(*ins*:*nfsb-mCherry*) fresh sperm with and without Met exposure. Sperm from Tg(*ins*:*nfsb-mCherry*) under control and diabetic conditions was cryopreserved and sperm motility and DNA integrity were evaluated.

### 4.3. Induction of Diabetes on Tg(ins:nfsb-mCherry) Zebrafish Model

Adult AB and Tg(*ins*:*nfsb-mCherry*) males of similar size were selected and separated into two groups, namely the control and diabetic treatment groups. Diabetes induction was performed with Met due to its efficacy in ablating β cells of the Tg(*ins*:*nfsb-mCherry*) line. Met was dissolved in system water at a final concentration of 10 mM with 0.5 mL/L of dimethyl sulfoxide (DMSO) by vigorous agitation. For the control treatment, the same conditions of water and fish housing were used with the exception of Met exposure. Males from both treatments were incubated at 28 °C in the dark for 24 h in glass tanks (14 fish/L of water). After the incubation, the males were returned to clean system water tanks. Zebrafish have regeneration ability; therefore, under transient diabetic conditions, maximum β-cell ablation is reached 3 days after exposure to the drug and pancreatic cells are fully regenerated 14 days after Met exposure [46]. Additionally, zebrafish males require 6 days to complete spermatogenesis [47]. Therefore, the diabetes induction methodology previously described was performed 7 days after the first Met exposure to ensure that all males had their spermatogenic cycle exposed to diabetic (or control) conditions.

### 4.4. Sperm Collection

Zebrafish were euthanized by hypothermal shock in ice water slurry at 2 ± 2 °C (monitored with a probe). The ice was removed from the slurry to avoid contact of the fish skin with the ice. This method avoids anesthetic interference with blood glucose [48] and facilitates fast euthanasia with respect to fish welfare [49,50]. In zebrafish, blood glucose levels rise 3 min after the exposure to stress [48]. Therefore, to avoid interference related to stress, all males in each tank were sacrificed within 3 min after the beginning of the tank manipulation. After the fish were properly sacrificed, the males were rinsed in phosphate-buffered saline (PBS) solution and cleaned with a paper towel to avoid sperm motility activation. Sperm was collected by abdominal massage using a glass capillary tube connected to a mouthpiece. Sperm was immediately diluted with 10 µL of sterilized and filtered (0.20 µm) Hank’s Balanced salt solution (HBSS) at 300 mOsm/kg [51] to prevent motility activation, in accordance with a previous study [52]. After sperm collection, the samples were maintained at 4 °C in the dark until quality analysis was performed (between 1 and 2 h after collection).

### 4.5. Pancreatic Fluorescence Observation

To confirm the diabetic condition of the zebrafish Tg(*ins*:*nfsb-mCherry*) males after sperm collection and to avoid blood glucose variations related to stress [48], males were separated 3 days prior to fish sampling into glass tanks (2 L of water) with 2 males of the same treatment per tank. Therefore, on the sperm sampling day, males of each tank could be sacrificed and sperm collected within 3 min, thus avoiding stress-related blood glucose increases that may have promoted biases in the glucose-related transcripts. Moreover, prior to sampling, males were fasted for 24 h to avoid differential blood glucose fluctuations related to food consumption. The observation of pancreatic fluorescence in Tg(*ins*:*nfsb-mCherry*) males was immediately performed. In adult fish, the observation of pancreatic fluorescence was impaired due to the high muscular density surrounding the tissues. Therefore, each fish was dissected and fluorescence was observed under a MZ 7.5 fluorescence stereomicroscope (Leica Microsystems GmbH, Wetzlar, Germany) equipped with a green light filter (λex = 530–560 nm and λem = 580 nm) coupled to a black and white F-View II camera (Olympus, Hamburg, Germany), which was controlled by Cell^F v2.7 software (Olympus Soft Imaging, Münster, Germany).

### 4.6. RNA Extraction and Complementary DNA Synthesis

Total RNA was extracted from sperm pools of: control groups AB (*n* = 4) and Tg(*ins*:*nfsb-mCherry*) (*n* = 7) and Met-treated groups AB (*n* = 4) and Tg(*ins*:*nfsb-mCherry*) (*n* = 7). Each pool contained sperm from 5 males. The sperm sample of each male was collected, diluted in 10 µL of phosphate-buffered buffer (PBS), and added to NZYol reagent (NZYTech, Lisbon, Portugal), according to the manufacturer’s specifications. DNAse treatment was performed using RQ1 RNase-Free DNase product (Promega, Madison, WI, USA) to remove genomic DNA contamination. The purity of the RNA samples was evaluated. The concentration and purity of the total RNA samples were evaluated using a NanoDrop 1000 (Thermo Fisher Scientific, Waltham, MA, USA) and the 260/280 ratios were 1.8–2.0. The integrity of the obtained RNA was assessed through Experion RNA analysis (Biorad, Hercules, CA, USA). Complementary DNA (cDNA) was synthesized from 500 ng of the total RNA using the M-MLV reverse transcriptase kit (Thermo Fisher Scientific, Waltham, MA, USA) with an oligo (dT) primer following the manufacturer’s protocol. The reverse transcription conditions were 37 °C for 1 h followed by 70 °C for 15 min, and the samples were stored at −20 °C until further analysis.

### 4.7. Quantitative Real-Time Polymerase Chain Reaction (qPCR)

Relevant genes affected by diabetes were selected for analysis in zebrafish spermatozoa, namely *insa*, *insra*, and *slc2a2.* The qPCR primers were designed using Perl Primer software (open-source PCR primer design). The primer nucleotide sequences are described in Appendix A. During the preliminary trials, genes related to oxidative stress (*hif1a* and *caspa3*) and a gene downregulated in the sperm of diabetes patients (*b4galt2*) were analyzed in zebrafish sperm under all treatments (Appendix A).

The real-time PCR (qPCR) conditions were optimized for the different primers (Appendix A). The amplification was monitored and analyzed by the intercalation of the fluorescent dye, SYBR Green, to double-stranded DNA. Reaction mixtures (20 μL of total volume) contained template cDNA (100 ng cDNA), SYBR Green PCR Master Mix (10 μL), and 10 µM each of the forward and reverse primers (0.8 μL). Quantitative PCR was initiated with a pre-incubation phase of 30 s at 95 °C, followed by 40 cycles with a denaturation phase of at 95 °C for 10 s, and primer extension and annealing at 60–62 °C for 20 s. To check the specificity of the qPCR reactions, a dissociation curve analysis was also included with 0.5 °C increments from 60 to 95 °C. Quantitative PCR amplification and fluorescence detection were carried out in a StepOnePlus™ System (Applied Biosystems, Waltham, MA, USA) according to the guidelines provided. StepOnePlus™ Systems software v.2.0 was used to calculate threshold cycle values (Ct). The ef1α gene was used as the endogenous reference gene to correct for differences in reverse transcription efficiency and template quantity [53]. The mRNA levels were calculated as fold expression relative to the housekeeping gene, ef1α. Each sample was analyzed in duplicate and the results were expressed according to the method described by Bustin [54]. Relative changes in gene expression were quantified using the 2^−ΔΔCt^ method [55]. The linearity, detection range, and qPCR amplification efficiency of each primer were checked before proceeding with sample analysis.

### 4.8. Sperm Plasma Membrane Viability Analysis

Sperm membrane viability was assessed through flow cytometry using SYBR 14 (Invitrogen, Thermo Fisher Scientific, Waltham, Massachusetts, USA) and propidium iodide (PI) (Sigma Aldrich, Spain) labeling, as described by Diogo et al. [44]. The plasma membrane viability of spermatozoa was evaluated in Tg(*ins*:*nfsb-mCherry*) males exposed to control (*n* = 5) and Met treatment (*n* = 6). SYBR 14 is a permeant nucleic acid dye that permeates the cellular plasma membrane and PI is a membrane impermeable dye; therefore, PI only labeled cells with disrupted membranes. Consequently, cells with disrupted membranes were labeled red from PI and viable cells were labeled green from SYBR 14 [56]. The SYBR 14 was diluted with 5 µL of stock solution added to 120 µL of sterilized and filtered HBSS, while PI was used undiluted. The pre-diluted sperm samples were re-diluted (1:300) in HBSS and each stain was added for a final concentration of 6.7 nM of SYBR 14 and 3 ng/mL of PI. The samples were incubated for 5 min in the dark at room temperature (21 to 25 ± 1 °C). Sperm plasma membrane viability analysis was performed using a flow cytometer (BD FACSCalibur™, Biosciences, Madrid, Spain). The flow cytometer settings were adjusted for the detection of SYBR 14 through a 530 nm bandpass filter (FL1) and PI was detected using a 670 nm long pass filter (FL3). Prior to the beginning of the experiments, the flow cytometer settings were adjusted for zebrafish sperm analysis using positive (100% dead cells) and negative (fresh sperm) controls. For the negative control, spermatozoa were exposed to cycles of freezing thawing [57]. A total of 5000–10,000 events were counted for each sample.

### 4.9. Sperm Cryopreservation and Thawing

For sperm cryopreservation, sperm samples from individual males were re-diluted in HBSS (1:2) and a final concentration of 10% of dimethylformamide. Sperm was cryopreserved in a final volume of 10 µL in each cryovial (low-temperature freezer vials; VWR^®^ International, Radnor, PA, USA). The sperm samples were cryopreserved in 2 mL cryovials with a controlled freezing rate of −10 °C/min in a programmable Assymptote EF600M biofreezer (Grant, Swindon, UK). The cryopreserved samples were then plunged into liquid nitrogen and stored in the cryobank. The samples were thawed in a 33 °C bath for 8 s and immediately analyzed.

### 4.10. Sperm Motility Analysis

Sperm motility analysis was evaluated in fresh sperm obtained from Tg(*ins*:*nfsb-mCherry*) males in the control (*n* = 5) and Met treatment (*n* = 8) groups and in cryopreserved sperm from the control (*n* = 5) and Met treatment (*n* = 10) groups. Sperm motility was evaluated using the computer-assisted sperm analysis (CASA) system (ISAS Integrated System for Semen Analysis, Proiser, Valencia, Spain) coupled to a phase contrast microscope (Nikon E-200, Nikon, Tokyo, Japan) with a x10 negative phase contrast objective. The images were captured with a Basler camera ISAS 782C camera (Proisier, Valencia, Spain) and processed with CASA software. The settings of the CASA system were adapted for this species [58]. Motility analysis was performed by placing 1 μL of pre-diluted sperm in a Mackler chamber and sperm motility was immediately activated with 5 μL of filtered (0.20 μm) and sterilized system water set at 28 °C. Sperm motility was characterized for 1 min each 10 s post-activation according to total motility (TM; %), progressive motility (PM; %), curvilinear velocity (VCL; μm/s), straight-line velocity (VSL; μm/s), and linearity (LIN; %). Only sperm samples with VCL > 10 μm/s were considered motile.

### 4.11. DNA Integrity Evaluation through Comet Assay

DNA integrity was evaluated in fresh sperm obtained from Tg(*ins*:*nfsb-mCherry*) males in the control (*n* = 4) and Met treatment (*n* = 8) groups and in cryopreserved sperm from the control (*n* = 5) and Met treatment (*n* = 6) groups. DNA fragmentation was evaluated through Comet assay methodology adapted from Reinardy et al. [59], with some modifications as previously described by Diogo et al. [44]. The fresh pre-diluted sperm (1 μL) or thawed sperm (3 μL) was diluted in 60 μL of low melting point agarose (0.5% in PBS). The samples diluted in low melting point agarose were distributed onto pre-coated slides with 0.5% agarose in PBS (dried overnight) and covered with a coverslip for 15 min at 4 °C. For the positive control, 2 μL of pre-diluted sperm was incubated with 2 μL of 100 μM H_2_O_2_ for 20 min at 4 °C to induce DNA fragmentation. The coverslip was removed, and the slides were incubated in lysis solution (2.5 M NaCl, 100 mM EDTA, 10 mM Tris, and 1% Triton X-100) for 1 h at 4 °C. Afterwards, the slides were placed in an alkaline electrophoresis solution (300 mM NaOH and 1 mM EDTA, pH 13) for 20 min to unwind the DNA. Electrophoresis was then performed for 20 min at 25 V and 280–300 mA. The slides were washed twice with neutralization solution (0.4 M Tris–HCl, pH 7.5) for 5 min and fixed in ethanol for 15 min. For sample visualization, the DNA present in each slide was labeled with 10 μL of PI (1 mg/mL) and immediately observed at 600× magnification under a fluorescence microscope (Olympus IX 81, Olympus, Tokyo, Japan) with blue excitation at 450–480 nm. Images were captured and recorded with a digital camera (F-view, Olympus, Tokyo, Japan) and processed with Cell^F image software (Olympus, Tokyo, Japan). At least 100 cells per slide were scored and further analyzed using Kinetic Imaging Komet 5.5 software (Andor Technology Ltd., Belfast, UK). DNA fragmentation was expressed in terms of DNA in tail (%).

### 4.12. Data Analysis

IBM SPSS Statistics 25.0 software (International Business Machines Corporation, Armonk, NY, USA) was used to perform statistical analysis. Data were expressed as means ± SD (standard deviation) and normalized by logarithmic or arcsine transformation when results were expressed as percentages. The genomic transcripts present in spermatozoa obtained by quantitative real-time PCR were evaluated through one-way ANOVA with the post hoc Student–Newman–Keuls (SNK) test (*p* < 0.05). Plasma membrane viability results were compared using the independent samples *t*-test (*p* < 0.05). A repeated measures ANOVA was applied for sperm motility analysis and each time post-activation was evaluated using the independent samples *t*-test for fresh and cryopreserved sperm (*p* < 0.05).

To evaluate whether sperm freezability was affected by the diabetic condition, fresh and cryopreserved sperm from untreated and Met-treated Tg(*ins*:*nfsb-mCherry*) males were evaluated according to sperm motility and DNA fragmentation parameters. Viability of the sperm plasma membrane was not used since the preliminary data analysis revealed that it was not useful in the comparison between fresh and cryopreserved samples in the present work. Due to the high number of variables related to sperm quality (4 DNA fragmentation parameters + 5 motility parameters x 6 post activation times = 34 sperm quality variables) measured for each sample, their degree of redundancy was investigated. Consequently, principal component analysis (PCA) was used to assess the possibility of aggregating all variables into a small number of components without significant loss of information. To check the sperm freezability of untreated (fresh *n* = 5; cryopreserved *n* = 4) and Met-treated (fresh *n* = 7; cryopreserved *n* = 9) Tg(*ins*:*nfsb-mCherry*) males, hierarchical cluster analysis was applied to the scores of PCA components. Ward’s method [60] was applied since this methodology allows the formation of hierarchical groups of mutually exclusive subsets, where each member of the group is maximally similar in relation to their inherent characteristics (i.e., sperm motility and DNA fragmentation). To apply Ward’s method, the squared Euclidean distance was fixed computationally. This methodology is an agglomerative hierarchical clustering procedure for classifying the homogeneity of samples according to a multivariate perspective. All the variables were considered according to treatment (control and diabetic) and sperm (fresh and cryopreserved) sample. This analysis is represented through a dendrogram in Appendix A.

## 5. Patents

No patents resulted from the work reported in this manuscript.

## Figures and Tables

**Figure 1 ijms-24-07035-f001:**
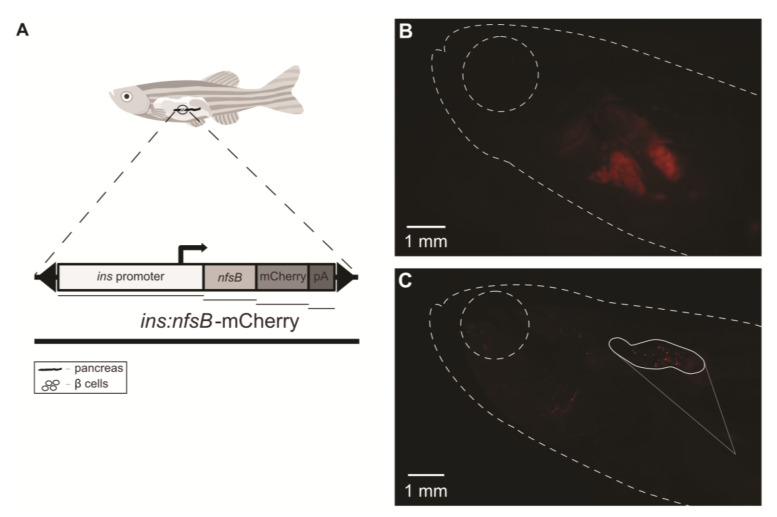
Confirmation of pancreatic β-cell ablation and onset of diabetes in zebrafish males. (**A**) Graphic representation of the construct inserted in the Tg(*ins*:*nfsb-mCherry*) genome. (**B**) Observation of pancreas with lack of fluorescence in Tg(*ins*:*nfsb-mCherry*) diabetic males, with evident pancreatic β-cell ablation and onset of diabetes. (**C**) Observation of pancreatic fluorescence in Tg(*ins*:*nfsb-mCherry*) control males after dissection, with the presence of live β cells (white area showing magnification of the pancreas).

**Figure 2 ijms-24-07035-f002:**
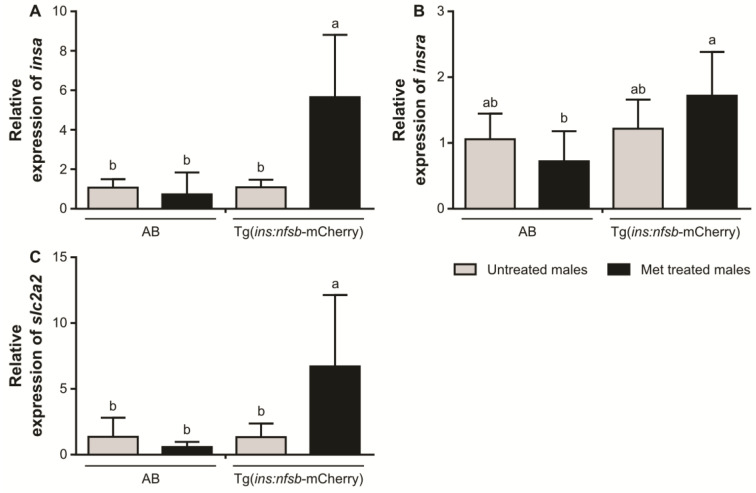
Gene expression of insulin metabolism markers in zebrafish sperm pools: (**A**) expression of *insa* in spermatozoa pools from AB males in control (*n* = 4) and metronidazole (Met) treatment (*n* = 4) groups and Tg(*ins*:*nfsb-mCherry*) in control (*n* = 5) and Met treatment (*n* = 7) groups; (**B**) expression of *insra* in spermatozoa pools from AB males in control (*n* = 4) and Met treatment (*n* = 4) groups and Tg(*ins*:*nfsb-mCherry*) in control (*n* = 7) and Met treatment (*n* = 7) groups; (**C**) expression of *slc2a2* in spermatozoa pools from AB males in control (*n* = 4) and Met treatment (*n* = 4) groups and Tg(*ins*:*nfsb-mCherry*) in control (*n* = 5) and Met treatment (*n* = 6) groups. The plotted values represent means ± SD. Different letters represent statistical differences (one-way ANOVA, post hoc SNK, *p* < 0.05).

**Figure 3 ijms-24-07035-f003:**
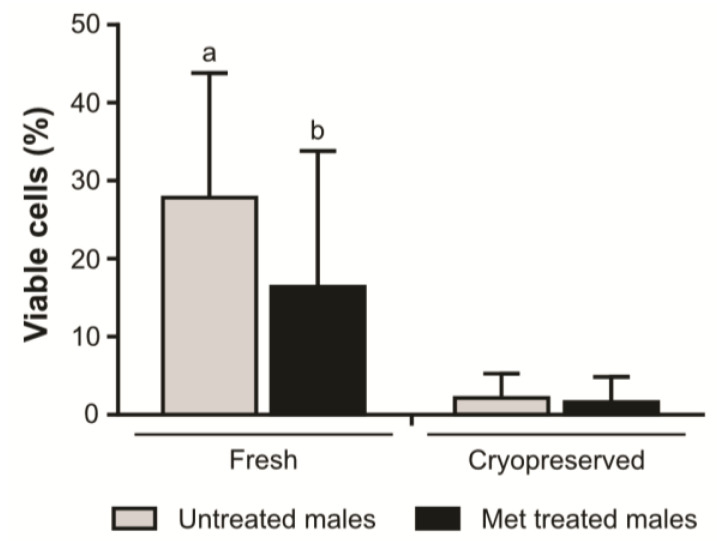
Spermatozoa plasma membrane viability of fresh and cryopreserved sperm from untreated Tg(*ins*:*nfsb-mCherry*) males (*n* = 5) and metronidazole (Met)-treated males (*n* = 6). The plotted values represent means ± SD. Different superscript letters represent statistical differences (independent samples *t*-test, *p* < 0.05).

**Figure 4 ijms-24-07035-f004:**
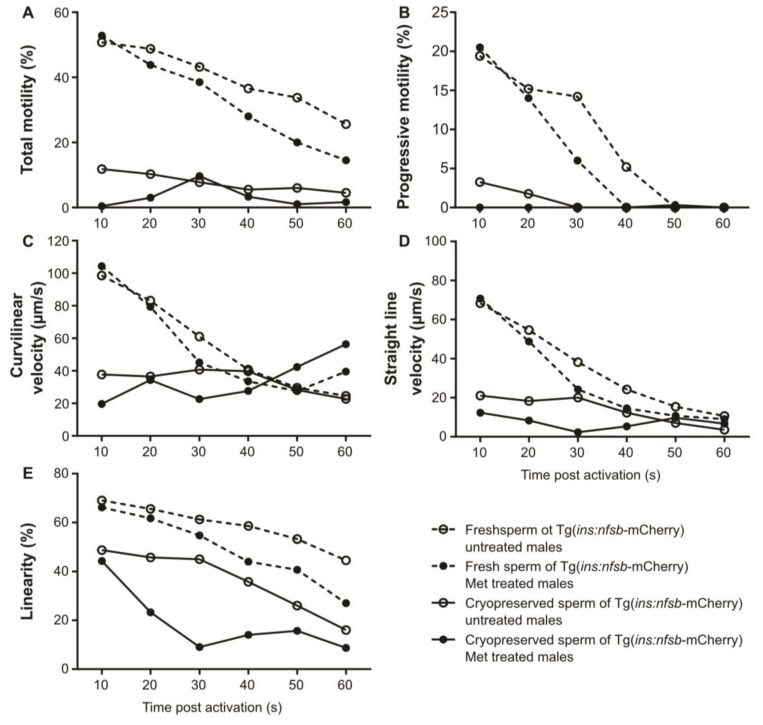
Sperm motility parameters of fresh sperm obtained from untreated Tg(*ins*:*nfsb-mCherry*) males (*n* = 5) and metronidazole (Met)-treated males (*n* = 8) and of cryopreserved sperm from untreated (*n* = 5) and Met-treated males (*n* = 10). Sperm was activated, and motility parameters were recorded every 10 s for 1 min in terms of: (**A**) total motility (%); (**B**) progressive motility (%); (**C**) curvilinear velocity (μm/s); (**D**) straight-line velocity (μm/s), and (**E**) linearity (%). The plotted values represent means. The dashed lines represent fresh sperm and the continuous lines represent cryopreserved sperm. Sperm from untreated males is represented with a white circle and sperm from Met-treated males is represented with a dark circle.

**Figure 5 ijms-24-07035-f005:**
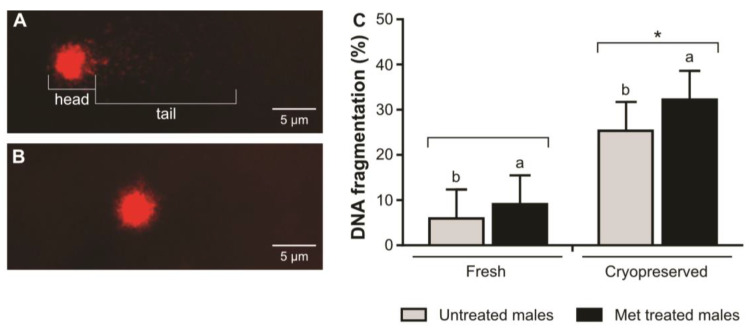
DNA integrity in zebrafish sperm: (**A**) comet with high DNA fragmentation; (**B**) comet with low DNA fragmentation; (**C**) DNA fragmentation of fresh sperm obtained from untreated Tg(*ins*:*nfsb-mCherry*) males (*n* = 4) and metronidazole (Met)-treated males (*n* = 8) and of cryopreserved sperm from untreated males (*n* = 5) and Met-treated males (*n* = 6). The plotted values represent means ± SD. Different letters represent statistical differences between treatments and the asterisk represents a statistical difference between fresh and cryopreserved sperm (independent samples *t*-test, *p* < 0.05).

**Figure 6 ijms-24-07035-f006:**
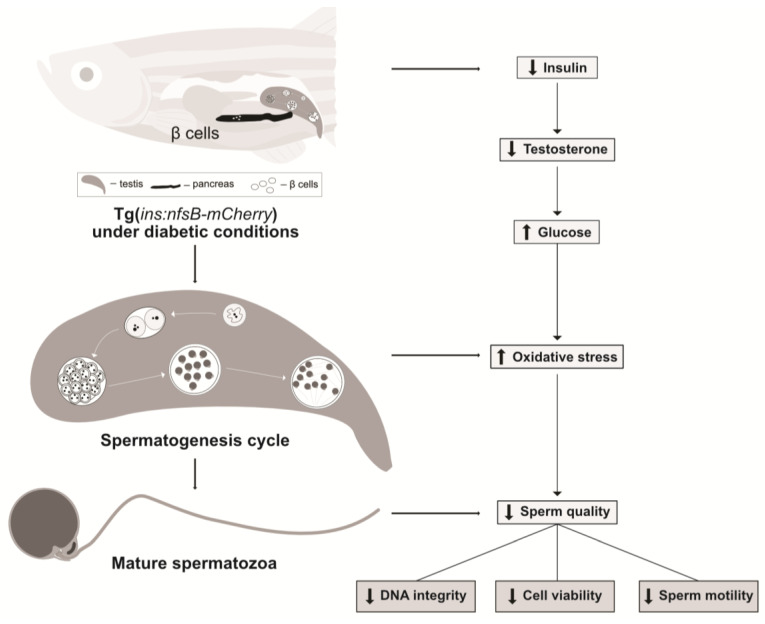
Hypothetic model of putative effects of diabetes on sperm quality in type I diabetes zebrafish strain [Tg(*ins*:*nfsb-mCherry*)].

**Table 1 ijms-24-07035-t001:** Statistical analysis of sperm motility parameters (data recorded for 1 min, each 10 s post-activation) in Tg(*ins*:*nfsb-mCherry*) zebrafish under control and diabetic conditions and their interactions (*p*-values).

Repeated Measures ANOVA	TM (%)	PM (%)	VCL (µm/s)	VSL (µm/s)	PM (%)
Treatment (control/diabetic)	0.017 *	0.050 *	0.023 *	0.018 *	0.006 *
Sperm (fresh/cryopreserved)	<0.001 *	<0.001 *	0.045 *	0.005 *	0.005 *
Treatment × sperm	0.530	0.165	0.717	0.943	0.999

Significant differences (repeated measures ANOVA, *p* < 0.05) are represented with an asterisk.

## Data Availability

The data presented in this study are available upon request from the corresponding author.

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
