# Peer review of "Type I Diabetes in Zebrafish Reduces Sperm Quality and Increases Insulin and Glucose Transporter Transcripts"

_ijms, 2023, doi:10.3390/ijms24087035_

Round 1
Reviewer 1 Report
Generally speaking, the study does not seem new to me because there is a lot of information indicating the effect of metabolic diseases such as diabetes on spermatogenesis and sperm quality. Diabetes and other metabolic diseases are known to induce reproductive damage. However, I think the study could be important for using zebrafish as a model for metabolic diseases.
There are some observations about it:
line 16: "the cause of diabetes is not known", this support is false, because it is already known that it is multifactorial. This same changes in the introduction.
Oxidative stress is an effect closely related to diabetes and DNA damage, so it would be necessary to include an assay such as the evaluation of ROS levels in spermatozoa due to diabetes.
Something more should be added to the discussion from other studies related to diabetes and reproductive damage.
Author Response
line 16: "the cause of diabetes is not known", this support is false, because it is already known that it is multifactorial. This same changes in the introduction.
The sentence in the manuscript was rephrased in order to characterize the onset of diabetes type I as polygenic.
Oxidative stress is an effect closely related to diabetes and DNA damage, so it would be necessary to include an assay such as the evaluation of ROS levels in spermatozoa due to diabetes.
During the experiment it was possible to perform a preliminary trial to test several relevant genes for the purpose of the present work in zebrafish sperm. Upon this trial the genes investigated in zebrafish sperm (of each treatment), presented upon the manuscript submission, were selected due to their relevance. However, it was possible to collect data in spermatozoa genes of the different zebrafish strains (AB and Tg(ins:nfsb-mCherry)). with both treatments (non-induced and with met treatment) related to oxidative stress namely caspase 3a and hif1a. Additionally, one gene highly downregulated in sperm from diabetes patients (b4galt2) was evaluated in the preliminary test. For the genes represented in the initial manuscript more biological replica were used. However, for the preliminary evaluation of these 3 genes sperm from only 2 males were used (with 4 technical replica each) and for that reason they were added to the supplementary data of the manuscript, since no statistical analysis can be applied to this data. Hypoxia-Inducible Factor-1 (HIF-1) HIF-1alfa is a gene related to cell survival in hypoxic conditions. The regulation of HIF-1 is a complex process and involves a number of molecules and pathways. Among these mechanisms a direct regulatory role of reactive oxygen species (ROS) on HIF-1 alpha subunit has received a great deal of attention (Shahrzad Movafagh, Sean Crook, Kim Vo 2015). Our data show a tendency for the reduction of hif1a under diabetic conditions and The Repression of hypoxia-inducible factor-1 is known to contribute to increased mitochondrial reactive oxygen species production in diabetes (Zheng et al 2022, reference number 37 in the manuscript).
Something more should be added to the discussion from other studies related to diabetes and reproductive damage.
Thank your for the suggestion, the recommendation was followed and the manuscript improved.
Reviewer 2 Report
The manuscript "Diabetes Type I in zebrafish reduces sperm quality and increases insulin and glucose transporters transcripts" by Diogo et al. discusses the detrimental effects of type I diabetes on reproduction, in particular on sperm motility and DNA integrity. The study aimed to investigate sperm quality and relevant genes to diabetes in spermatozoa of Tg(ins:nfsb-mCherry) zebrafish, a model for type I diabetes. The results showed that diabetic zebrafish had significantly higher quantities of insulin and glucose transporter transcripts compared to control, and their sperm had significantly lower motility, plasma membrane viability, and DNA integrity. The study validates the zebrafish model for type I diabetes research in germ cells.
Overall, the manuscript is very well elaborated and extensive. The language, style, grammar, syntax, and overall quality of the text are appropriate for a scientific study. some of the sentences of the introduction seem a bit off, though. It might be helpful to break down some of the longer sentences into multiple sentences. The introduction seems a bit lengthy, too.
The presentation of the results and their discussion is adequate. The methods are very well described and allow reproducibility. All these sections are fine as submitted.
Author Response
The manuscript "Diabetes Type I in zebrafish reduces sperm quality and increases insulin and glucose transporters transcripts" by Diogo et al. discusses the detrimental effects of type I diabetes on reproduction, in particular on sperm motility and DNA integrity. The study aimed to investigate sperm quality and relevant genes to diabetes in spermatozoa of Tg(ins:nfsb-mCherry) zebrafish, a model for type I diabetes. The results showed that diabetic zebrafish had significantly higher quantities of insulin and glucose transporter transcripts compared to control, and their sperm had significantly lower motility, plasma membrane viability, and DNA integrity. The study validates the zebrafish model for type I diabetes research in germ cells.
Overall, the manuscript is very well elaborated and extensive. The language, style, grammar, syntax, and overall quality of the text are appropriate for a scientific study. some of the sentences of the introduction seem a bit off, though. It might be helpful to break down some of the longer sentences into multiple sentences. The introduction seems a bit lengthy, too.
The authors thank the comments of the reviewer and the recommendations about the introduction of the manuscript were followed and the manuscript improved accordingly.
The presentation of the results and their discussion is adequate. The methods are very well described and allow reproducibility. All these sections are fine as submitted.
Round 2
Reviewer 1 Report
The comments were corrected, I think it should be accepted in its current form.